# Smart Restored by Learning Exercise Alleviates the Deterioration of Cognitive Function in Older Adults with Dementia—A Quasi-Experimental Research

**DOI:** 10.3390/ijerph16071270

**Published:** 2019-04-09

**Authors:** Chi-Fen Tseng, Shao-Huai Lee, Tsung-Cheng Hsieh, Ru-Ping Lee

**Affiliations:** 1Institute of Medical Sciences, Tzu Chi University, Hualien 97004, Taiwan; aa@ems.tcust.edu.tw (C.-F.T.); tchsieh@mail.tcu.edu.tw (T.-C.H.); 2Department of Nursing, Tzu Chi University of Science and Technology, Hualien 970, Taiwan; 3Department of Family Studies and Child Development, Shih Chien University, Taipei 10462, Taiwan; aminophyline@hotmail.com; 4Taiwan Smart Restored by Learning Exercise Development Association, Taipei 10696, Taiwan

**Keywords:** older adults with dementia, cognitive function, neuropsychiatric symptoms, frontal lobe function

## Abstract

Maintaining cognitive function is essential for older adults with dementia. The purpose of this study was to investigate the effectiveness of Smart Restored by Learning Exercise (SRLE) on cognitive functions, neuropsychiatric symptoms, and frontal lobe functions in elderly people with dementia. A total of 68 older adults with dementia participated in this study. A quasi-experimental design was used, and convenience sampling and assignment approaches were adopted to select the participants for experimental and control groups. The experimental group engaged in SRLE for 6 months. The control group received routine care without SRLE. The participants’ cognitive function, neuropsychiatric symptoms, and frontal lobe function at baseline were evaluated using the Mini-Mental State Examination (MMSE), Neuropsychiatry Inventory (NPI), and Frontal Assessment Battery (FAB), respectively, in month 3 and month 6. The Group by Time interaction was statistically significant for MMSE and FAB scores, which indicated the different group effects between months 3 and 6. The results also showed that the improvement of MMSE, NPI, and FAB scores in the SRLE group were significantly better than the control group (t = −5.99~4.90, *p* < 0.001) at both months 3 and 6. In conclusion, long-term facilities may provide residents with SRLE every day to prevent a decline in the residents’ levels of cognitive function.

## 1. Introduction

Dementia is a common degenerative disease among older adults. It is a syndrome characterized by slow degeneration, often accompanied by cognitive, emotional, behavioral, and psychological symptoms. These symptoms often lead to poor quality of self-care, fatigue of family members serving as caregivers, decreased quality of life, high mortality, and increased costs to the healthcare system [1,2]. In particular, patients with behavioral and psychological symptoms of dementia (BPSD) often cause a substantial burden on caregivers [3,4]. Behavioral and psychological symptoms of dementia are composed of a wide range of symptoms, including agitation, aggression, calling out repeatedly, sleep disturbance, wandering, and apathy [5]. The treatment strategies for BPSD include various pharmacological approaches such as antipsychotics, antidepressants, mood stabilizers, and cognitive enhancers. In a systematic review of two meta-analyses and two additional RCT studies, the results showed that there is no clear evidence for efficacy of conventional antipsychotic agents on several BPSD [6]. However, there is a meta-analysis that demonstrated the significant efficacy of atypical antipsychotics on psychiatric symptoms and cognitive functions compared to placebo [7]. Because drugs for dementia treatment exhibit limited effectiveness and cause numerous side effects, behavior therapy with drugs has become an alternative to drug treatment. In behavior therapy, operant conditioning and behavioral reinforcement or extinction are used to correct problematic behaviors. Standard measures include environmental changes, sound (including music) therapy [8,9], phototherapy, group activities, animal-assisted therapy, aromatherapy, massage therapy, reminiscence therapy, and physical activity [10]. These measures can be used to help individuals resolve personal conflicts and enhance self-understanding. They are also effective for the management of agitated behavior in older adults with dementia [9]. Furthermore, these strategies can be used to enhance the self-esteem, self-identification, and problem-solving abilities of older adults with dementia [11]. However, these measures are not effective for alleviating and stabilizing the neuropsychiatric symptoms caused by the decline of cognitive function. An approach for alleviating or reducing brain necrosis through brain cell activation to delay aging could offer hope to dementia patients and their family members.

Frontal lobe degeneration is significantly correlated with cognitive functions, such as memory impairment, declines in executive functions, and aphasia [12,13]. Decreased blood flow in the frontal, parietal, and temporal cortices of patients with dementia has been reported [14,15]. However, studies that have employed MRI technology for assessments have confirmed that oral reading and simple calculations can increase blood flow in the prefrontal cortex and parietal lobes, which can improve or maintain patients’ levels of cognitive function [16]. Uchida conducted a randomized experiment on healthy older adults aged 70 to 86 years. The participants were asked to read aloud and perform simple calculations for 6 months. The results indicated that these measures promoted and maintained the cognitive functions of the older adults in the experimental group. By contrast, the cognitive function of the participants in the control group degenerated [17]. This observation reflected that, without intervention through cognitive rehabilitation, the cognitive function of older adults may gradually degenerate, possibly resulting in severe disabilities. Studies have demonstrated that oral reading and the performance of simple calculations can slow the progression of cognitive decline, increase blood flow in the frontal lobe, and improve frontal lobe function [13,14]. Based on the principles of oral reading and performance of simple calculations proposed by Kawashima, Smart Restored by Learning Exercise (SRLE) was deployed. The participants in this study were asked to read aloud in Mandarin and to perform arithmetic calculations based on materials corresponding with third and fourth grade elementary school levels of difficulty. The intervention measures were conducted 4 to 5 days per week for 6 months. The purpose of this intervention was to explore related effects on cognitive functions, neuropsychiatric symptoms, and improvement or maintenance of frontal lobe function in the older adults with dementia. In the present study, the effectiveness of the SRLE program for reducing the decline in cognitive function was investigated in older adults with dementia in long-term care facilities. This study also investigated whether the SRLE could reduce the participants’ BPSD and improve or maintain their frontal lobe function.

## 2. Materials and Methods

### 2.1. Participants

The participants in this study were older adults who had received diagnoses of dementia and who resided in a veterans’ home. Leaflets were distributed to recruit participants in the veterans’ home. According to the inclusion criteria, a total of 68 older adults with dementia participated in the study. The inclusion used to determine whether recruited individuals were enrolled in the study criteria were older adults diagnosed with mild to severe dementia (Mini-Mental Status Examination [MMSE] score < 25), and patients with dementia and encephalatrophy diagnosed by clinical neuromedical experts. Older adults diagnosed as being without dementia and aphasia, and those individuals who took Cholrpromazine and cognitive enhancers, were excluded. This study was approved by Hualien Tzu Chi Hospital Research Ethics Committee (REC No: IRB 102-170).

### 2.2. Study Design

A quasi-experimental study was conducted using convenience sampling and assignment. Accordingly, participants were assigned to an experimental group (SRLE) or a control group. In the veterans’ home, the older adults with dementia resided on the first to third floors. To avoid mutual influence, 33 participants from the first floor were assigned to the control group, and 35 participants from the second and third floors were assigned to the experimental group. For the experimental group, the SRLE was applied as an intervention measure and routine care was provided. The control group only received routine care.

Three measurement tools were used in this study. The MMSE [18] was used to assess the cognitive functioning of the older adults with dementia. The Neuropsychiatric Inventory (NPI) [19] was used to assess neuropsychiatric symptoms. Finally, the Frontal Assessment Battery (FAB) [20] was adopted to assess frontal lobe functioning. Baseline measurements were performed in the week before the intervention was deployed. Additional measurements were conducted in the third and sixth months of the intervention. The MMSE measure exhibits favorable validity and reliability [21] and it is used to test the cognitive functioning of patients with brain injury. The test is comprised of eight subtests that assess respondents’ cognition functions, including orientation (spatial and time), attention, registration and recall, calculation, writing, language, and visuospatial construction. High scores represent high levels of cognitive functioning. Chinese–Taiwanese MMSE16 was established with a Cronbach’s alpha coefficient of 0.8 in this study.

The NPI exhibits acceptable reliability and validity [22] and covers 12 behavioral symptoms: delusions, hallucinations, agitation or aggression, depression or dysphoria, anxiety, elation or euphoria, apathy or indifference, disinhibition, irritability or lability, bizarre behavior, sleep disturbances, and appetite or eating disorders. High scores indicate great severity of the corresponding symptom. The present study used the Chinese version of the NPI [19], and the Cronbach’s alpha coefficient was 0.91.

The FAB was adopted for evaluations of frontal lobe functioning. The FAB showed good validity and reliability [23]. The FAB consists of six subsets: conceptualization, mental flexibility, motor programming, and sensitivity to interference, inhibitory control, and environmental autonomy. High scores reflected high levels of frontal lobe functioning. The Taiwanese version of the Frontal Assessment Battery Cronbach’s alpha coefficient was 0.85 in this study [20].

### 2.3. Intervention

#### 2.3.1. Smart Restored by Learning Exercise (SRLE) Task

The SRLE used in this study was comprised of two tasks: arithmetic and mandarin Chinese. (1) Arithmetic: Simple arithmetic materials from third and fourth grade elementary school textbooks were used. The content ranged from simple addition and subtraction to relatively difficult three-digit addition and subtraction. (2) Mandarin Chinese: The participants were asked to systematically read the Mandarin Chinese text aloud, and the materials were selected from third and fourth grade elementary school textbooks. The reading materials involved sentences of varying difficulty, including simple, medium, and complex. The difficulty of sentences was categorized according to three levels.

#### 2.3.2. Implementation Methods

The implementation was an individualized strategy instead of a group intervention. The materials from the Mandarin Chinese and mathematics textbooks were printed on A4 size paper. Each participant in the SRLE group was assigned appropriate difficulty levels and materials and evaluated according to their individual levels. There were three levels of difficulty, the simple, medium, and complex levels. Each participant had been tested with a simple-level material first by the intervener, if all answers were correct, then give a medium-level material, if there was an incorrect answer, or the participant had to think for a long time in the answering, simple-level material would be assigned in the following experimental period, namely, the difficulty level was based on whether the material could be easily read and the calculation performed without errors by the participant. While the simple and easy-to-complete materials can promote the blood flow to the frontal lobe, on the contrary, the difficult materials reduce the blood flow to the frontal lobe. This preparation facilitated the smooth engagement of the participants with their interventional materials. Each participant in the SRLE group was required to complete an assignment of 4 to 5 sheets of mathematics problems (A4 size double-sided) and 4 to 5 sheets of Mandarin Chinese reading from Monday to Friday. Approximately 20 to 30 min were allotted for the participants to study one sheet each of Mandarin Chinese and mathematics. Each week, the completed Mandarin Chinese and mathematics assignments were evaluated by the unit staff or the researchers. If the participants encountered problems, they were assisted by the staff in the unit or the researchers.

#### 2.3.3. Implementation Period and Principles

The intervention measures were conducted in the SRLE group 4 to 5 days per week for 6 consecutive months. There was only one intervener to perform the intervention for the 35 participants. Data were collected before deployment, 3 months into the intervention, and after the intervention period by the intervener. Timely and positive feedback was provided to the participants.

### 2.4. Statistical Analysis

The descriptive statistics including mean, standard deviation, frequency, and percentage were presented for continuous and categorical variables. The demographic variables, including age, onset, education level, MMSE, NPI, and FAB baselines of the two groups, were analyzed using the Wilcoxon ranked sum test. The Wilcoxon signed rank test was performed for evaluating the improvement (change from months 3 and 6 from baseline) of each score within the SRLE and control groups. The generalized estimating equation (GEE) with the covariates of treatment group, time (baseline, months 3, 6), and their interaction terms (group-by-time) was conducted for evaluating the treatment effect at 3 and 6 months after treatment. Statistical significance was set *p* < 0.05 for all the comparisons.

## 3. Results

A total of 68 older male adults with dementia participated in the present study: 35 adults in the SRLE group and 33 adults in the control group. The SRLE group and the control group were not significantly deferent in demographics (Table 1). All of the participants were male veterans, and their education levels were mostly junior high school and lower.

The average MMSE pretest scores were lower for the SRLE group (M = 14.19, SD = 4.53) than for the control group (M = 13.29, SD = 7.67). These scores indicated that the older adults who participated in this study exhibited moderate cognitive impairment. The BPSD measure was assessed using the NPI. The average score for the SRLE group was 10.23 (SD = 11.66), and the average score for the control group was 9.66 (SD = 14.66). This result indicated that the participants exhibited moderate neuropsychiatric symptoms and caused a moderate level of distress for their caregivers. The FAB was used to assess frontal lobe functioning. The average score for the SRLE group was 5.45 (SD = 3.02), and that for the control group was 4.75 (SD = 4.11). The level of frontal lobe functioning was moderate to low. Except for a difference for age, no significant differences were found in the demographic variables and scores for the three scales between the two groups.

Regarding average MMSE score for cognitive functioning on the posttest, after 3 months, the score of the SRLE group increased from 14.19 (SD = 4.53) to 18.13 (SD = 5.83), and it further increased to 19.29 after 6 months (SD = 5.11). These results indicated a significant improvement in cognitive functioning, whereas the scores of the control group indicated a decreasing trend (Table 2). 

The score of neuropsychiatric symptoms for the SRLE group decreased substantially from 10.23 (SD = 11.66) to 2.77 (SD = 5.22) after 3 months, and then slightly decreased to 2.03 (SD = 4.25) after 6 months. Conversely, the scores for the control group increased. Therefore, the neuropsychiatric symptoms improved favorably for participants in the SRLE group than for participants in the control group. The FAB score for frontal lobe functioning increased from 5.45 (SD = 3.02) to 7.87 (SD = 3.69) after 3 months, and it further increased to 10.00 (SD = 3.94) after 6 months. The scores for the control group decreased. 

The GEE measure was used to analyze the effectiveness of the intervention measures. The statistically significant effect of Group by T1 (MMSE: B = 5.64, *p* < 0.001; NPI: B = −10.66, *p* < 0.001; FAB: B = 3.86, *p* < 0.001) and Group by T2 (MMSE: B = 6.88, *p* < 0.001; NPI: B = −12.28, *p* < 0.001; FAB: B = 6.00, *p* < 0.001) for all three outcome scores showed the effectiveness of SRLE at both month 3 and month 6 (Table 3). Furthermore, the increases of the corresponding B of the interaction effects from T1 (month 3) to T2 (month 6) indicated the effects of SRLE increase with time. 

## 4. Discussion

The results of this study demonstrated that the performance of simple reading aloud and calculations can effectively enhance cognitive functioning, mitigate BPSD, and improve frontal lobe functioning of older adults with dementia. Additionally, the difference between the SRLE group and control group increased with the length of intervention employment. The intervention measures significantly improved the cognitive and frontal lobe function levels of the participants in the SRLE group; conversely, the control group only maintained or demonstrated decreases in scores.

Reading aloud and simple calculations were used for healthy older adults and older adults with dementia. Previous randomized experiments have demonstrated that, after 6 months of consecutive training, similar intervention measures promoted and maintained the cognitive function of the older adults in the experimental group. The participants in the control group exhibited a declining trend in cognitive functioning [17,24]. This conclusion was consistent with the results of the present study. In a study conducted by Kawashima, older adults with Alzheimer-type dementia also participated in reading and simple calculation programs for 2 to 6 days every week for 6 consecutive months. The participants’ cognitive functioning and frontal lobe functioning significantly improved [16]. In the present study, the dementia type of most of the participants was vascular dementia, and significant differences in patients’ levels of cognitive functioning were evident after 3 months of the intervention. Patients with Alzheimer-type dementia may require longer training times for differences to become evident. Mahncke et al. [25] used computers to train the auditory discrimination and memory of older adults in a community. The experimental group received 1 hour of training every day, 5 days per week for 8 to 10 weeks. The training significantly enhanced the participants’ levels of cognitive functioning [25]. Therefore, brain training plans should be followed on an on-going basis, and healthy older adults should participate in cognitive training to maintain cognitive functioning and, thereby, prevent the development of dementia. In future research, 3-month training plans for healthy older adults may be conducted to investigate the effect of the intervention measures. In addition, future research is required to evaluate computerized version of SRLE as smartphone innovation was found to be beneficial to dementia patients and their caregivers [26].

In contrast to the study conducted by Kawashima, which indicated that relatively difficult reading texts and problems for calculation reduced blood flow in the frontal, parietal, and temporal lobes of the brain in older adults [14]. Therefore, in related intervention processes, the language and the arithmetic ability of each older adult with dementia should first be examined, and reading and calculation materials of appropriate difficulty should be assigned accordingly. Excessively difficult materials may incite older adults’ unwillingness to participate in training and may even cause them to become angry and melancholic. Therefore, materials should be designed based on three categories of content difficulty—difficult, moderate, and simple—to enable the individualization of intervention measures. Previous studies related to the effect of reading aloud and simple arithmetic training on older adults have indicated that the emotional and behavioral performance of older adults with dementia significantly improved after 7 weeks of training. The intervention measures were particularly effective for reducing hallucinations, depression, indifference, irritability, strange behaviors, and sleep disorders [27]. These results were consistent with the ones in the present study, in which participants’ scores for neuropsychiatric symptoms were significantly reduced after the individuals had participated in the intervention. Additionally, the agitation and aggression of older adults with dementia substantially decreased after 3 months of participation in the intervention. Numerous studies have proposed that increased cognitive functioning can increase control of sensory perception and thereby reduce BPSD, such as depression. Learning therapy increases cognitive functioning and has been reported as effective for alleviating BPSD in older adults with dementia [28,29]. These results are consistent with those reported in the present study. An empirical study indicated that the current non-drug therapy for rehabilitating brain activity involved facilitating interactions between older adults with dementia and other people in pleasant situations wherein praise is used to strengthen learning motivation and thereby encourage the use of residual brain function [29]. However, there were some limitations to the present study. The participants were residents of a veterans’ home, which is a long-term care facility for veterans. Accordingly, the participants were all male individuals, and most of the veterans were over the age of 80 years old. In addition, the data collector was the same as the intervener. This flaw may impact on post-test data and research results’ inference. The current study is potentially biased, because participants were not randomly assigned to the experimental group or control group. Therefore, further study would benefit from using RCT design and increasing the number of participant and older female adults with dementia.

## 5. Conclusions

The present study explained that the SRLE program potentially reduces the decline in cognitive functioning and frontal lobe functions were explored in dementia elders. Our findings showed that SRLE could reduce the participants’ neuropsychiatric symptoms. 

## Figures and Tables

**Table 1 ijerph-16-01270-t001:** Demographics of the participants (*n* = 68).

Variables	SRLE Group (*n* = 35) Mean (SD)	Control Group (*n* = 33) Mean (SD)	*Z*	*p*
Age	86.4 (7.09)	89.0 (5.27)	1.688	0.409
Education	8.5 (3.60)	7.6 (2.76)	−1.115	0.350
Onset	2.14 (1.14)	2.33 (1.49)	0.593	0.658
MMSE	14.19 (4.53)	13.29 (7.67)	−0.239	0.716
NPI	10.23 (11.66)	9.66 (14.66)	0.041	0.651
FAB	5.45 (3.02)	4.75 (4.11)	−0.469	0.557

Age, education, onset, MMSE, NPI, FAB: Wilcoxon ranked sum test.

**Table 2 ijerph-16-01270-t002:** Scores of MMSE, NPI and FAB by group and time (*n* = 68).

	SRLE Group (*n* = 35)				Control Group (*n* = 33)			
Time Points	Each Time	Change from Baseline	*Z*	*p*	Each Time	Change from Baseline	*Z*	*p*
	Mean (SD)	Mean (SD)			Mean (SD)	Mean (SD)		
MMSE								
Baseline	14.19 (4.53)				13.29 (7.67)			
3 month	18.13 (5.83)	3.94 (4.49)	4.08	<0.001	11.18 (6.59)	−2.11 (3.34)	−3.33	0.003
6 month	19.29 (5.11)	5.10 (4.26)	4.63	<0.001	10.48 (7.02)	−2.81 (3.77)	−3.26	0.004
NPI								
Baseline	10.23 (11.66)				9.66 (14.66)			
3 month	2.77 (5.22)	−7.46 (12.04)	−3.26	<0.001	12.83 (14.81)	3.17 (6.60)	2.33	0.020
6 month	2.03 (4.25)	−8.20 (11.18)	−3.96	<0.001	14.22 (15.86)	4.56 (6.79)	2.70	0.002
FAB								
Baseline	5.45 (3.02)				4.75 (4.11)			
3 month	7.87 (3.69)	2.42 (2.93)	3.56	<0.001	3.96 (3.79)	−0.79 (1.26)	−2.70	0.007
6 month	10.00 (3.94)	4.55 (3.10)	4.51	<0.001	3.81 (4.28)	−0.94 (1.92)	−2.50	0.012

*p* value is obtained based on the Wilcoxon rank-sum test.

**Table 3 ijerph-16-01270-t003:** Effects of the SRLE on patient’s MMSE, NPI and FAB in the post-intervention and 3-month follow-up (*n* = 68).

**MMSE**			**95%wald CI**			
	***B***	**SE**	**Lower**	**Ppper**	**Wald x^2^**	***p***
intercept	13.25	1.29	10.71	15.78	104.75	<0.001
Group	1.259	-	-	-	-	<0.001
T_1_	−1.82	0.77	−3.36	−0.31	5.59	0.018
T_2_	−1.89	0.80	−3.47	−0.33	5.64	0.018
Group*T_1_	5.64	-	-	-	-	<0.001
Group*T_2_	6.88	-	-	-	-	<0.001
Scale	38.34					
**NPI**			**95%wald CI**			
	***B***	**SE**	**lower**	**upper**	**wald x^2^**	***p***
intercept	9.09	2.36	4.47	13.71	14.89	<0.001
Group	0.57	3.01	−5.34	6.48	0.04	0.851
T_1_	3.47	1.40	0.73	6.22	6.14	0.001
T_2_	4.32	1.43	1.52	7.12	9.12	0.003
Group*T_1_	−10.66	2.46	−15.48	−5.84	18.77	<0.001
Group*T_2_	−12.28	2.36	−16.90	−7.66	27.17	<0.001
Scale	135.80					
**FAB**			**95%wald CI**			
	***B***	**SE**	**lower**	**upper**	**wald x^2^**	***p***
intercept	4.58	0.69	3.23	5.92	44.53	<0.001
Group	0.17	0.85	−1.49	1.82	0.04	0.844
T_1_	−0.76	0.22	−1.19	−0.32	11.49	0.001
T_2_	−0.77	0.35	−1.45	−0.09	4.96	0.026
Group*T_1_	3.86	0.64	2.61	5.11	36.49	<0.001
Group*T_2_	6.00	0.68	4.66	7.34	76.83	<0.001
Scale	14.296					

T_1_: Baseline and post-intervention (3-month) comparison. T_2_: Baseline and 6-month follow-up comparison. Group*time: Reference group (control group*baseline). -: not applicable.

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
