# Peer review of "Smart Restored by Learning Exercise Alleviates the Deterioration of Cognitive Function in Older Adults with Dementia—A Quasi-Experimental Research"

_ijerph, 2019, doi:10.3390/ijerph16071270_

Round 1

Reviewer 1 Report

Journal: International Journal of Environmental Research and Public Health

 Manuscript title: Smart Restored by Learning Exercise Alleviates the Deterioration of Cognitive Function in Older Adults with Dementia

Comments:

Title:  Please add research design in title

Abstract:

Line 23 to 26: The results showed the experimental group had significantly improved MMSE, FAB and NPI score than control group. Authors didn’t describe the interaction effect (group * time).

Line 24 and 25: statistical parameter is B, not Beta (β) for GEE.

Introduction

Please add descriptions of medication on BPSD and cognitive function.

Methods:

The experimental group has 36 participants. Was the experimental group was divided into small groups? Then, how many groups were there? How many interveners do you have? How did you control the fidelity between interveners?

Who were the data collectors? Please report the inter-rater reliability. Did you train the data collectors?

Authors’ statements (Line 127-128): Each participant in the experimental group was assigned appropriate difficulty levels and materials and evaluated according to their individual levels. How did you define the level? Who judged participants’ level?

Data analysis: please add Chlorpromazine (100mg) and cognitive enhancer in GEE model. Because antipsychotics impact on BPSD and cognitive enhancer drugs improve cognitive function, you should add Chlorpromazine (100mg) and cognitive enhancer drugs into GEE model.

Please report effect size formula.

The GEE suggested SRLE effectiveness enough. The assumptions of ANCOVA are Normal distribution, equal variance and so on. Did you examine the assumptions of ANCOVA? To my opinion, keep GEE and delete the ANCOVA part.

Please change age and duration of onset from categorical variables to continuous variables.

Please report IRB approve number.

Results

Please report effect sizes for SRLE on outcome indicators.

Table 1, please put age and duration of onset as continuous variables. The reason is: For 2 * 2table, it is generally recommended that there should be a minimum ‘expected’ frequency of 10 in each cell. If not, Fisher’s exact test should be used. No cell should contain zero observations.

Please change the p = 0.000 to p < .001, and report the third digit after the decimal point for all p values. It is not suitable for taking p values from third digit after the decimal point to rounding to two digit after the decimal point.

Please combine Table 3 and Table 4 and report parameters of control variables and add B(SE) on the GEE model tables. The GEE model should report Group* T1 (EG vs. CG and baseline vs 3-month) and Group * T2 (EG vs. CG and baseline vs 6-month) interaction effect.

 Discussion

Please particularly attend to keeping the discussion of the paper and the conclusions within the bounds of our findings. That is, be careful not to overstate the significance or strength of the findings given the research design.

Not sure that it is appropriate to conclude that the SRLE may significantly enhance the cognitive function, improve the frontal lobe function, and reduce the BPSD in male adults with dementia aged over 65 years because of the research design limitations of the study, particularly potential bias due to authors assigned participants into experimental group or control group. It may be more appropriate to conclude that the intervention has potential and further study would be beneficial using RCT designs.

Author Response

Point 1: Title:  Please add research design in title

Response 1: The research design “A quaxi-experimental research” has been added in title.

Point 2: Abstract:

2.1. Line 23 to 26: The results showed the experimental group had significantly improved MMSE, FAB and NPI score than control group. Authors didn’t describe the interaction effect (group * time).

Response 2.1: Thank you very much for the comments. The description about the interaction effect of group-by-time has been added in the abstract. Please refer to Line 23-27 for the revision.

2.2. Line 24 and 25: statistical parameter is B, not Beta (β) for GEE.

Response 2.2: Thank you for your reminding. The sentences have been changed as the description about the interaction effect of group-by-time. (Line 23-27)

Point 3: Introduction

Please add descriptions of medication on BPSD and cognitive function.

Response 3: Thank you for your kindly advices on this study. Your suggestions make our manuscript more solid and readable. We have added some descriptions as follows. “BPSD are composed of wide-ranging symptoms such as agitation, aggression, calling out repeatedly, sleep disturbance, wandering, and apathy [5]. The treatment strategies for BPSD include various pharmacological approaches such as antipsychotics, antidepressants, mood stabilizers, and cognitive enhancers. In a systematic review of two meta-analysis and two additional RCT studies, the results showed that there is no clear evidence for efficacy of conventional antipsychotic agents on several BPSD. However, there is a meta-analysis demonstrated a significant efficacy of atypical antipsychotics on psychiatric symptoms and cognitive functions compared to placebo [6].” (Line 41-48)

Point 4: Methods:

4.1. The experimental group has 36 participants. Was the experimental group was divided into small groups? Then, how many groups were there? How many interveners do you have? How did you control the fidelity between interveners?

Response 4.1: The experimental group was not divided into small groups. There was only one intervener to perform the intervention for the 35 participants. We added it on Line 159.

4.2. Who were the data collectors? Please report the inter-rater reliability. Did you train the data collectors?

Response 4.2: There was only one data collector. Data were collected after the intervention by the intervener. (Line 161)

4.3. Authors’ statements (Line 127-128): Each participant in the experimental group was assigned appropriate difficulty levels and materials and evaluated according to their individual levels. How did you define the level? Who judged participants’ level?

Response 4.3: There were three levels of difficulty, the simple, medium, and complex levels. Each participant had been tested with a simple-level material first by the intervener, if all answers were correct, then give a medium-level material, if there was a mistaken answer or the participant had to think for a long time in the answering, a simple-level material would be assigned in the following experimental period, namely, the difficulty level based on whether the material can be easily read and the calculation without errors by the participant. Because the simple and easy-to-complete materials can promote the blood flow of the frontal lobe, on the contrary, the difficult materials reduce the blood flow of the frontal lobe.  We added this paragraph on Line 142-149.

4.4. Data analysis: please add Chlorpromazine (100mg) and cognitive enhancer in GEE model. Because antipsychotics impact on BPSD and cognitive enhancer drugs improve cognitive function, you should add Chlorpromazine (100mg) and cognitive enhancer drugs into GEE model.

Response 4.4: Thank you for your suggestion. Because Chlorpromazine and cognitive enhancers do impact on cognition, so, in the study design, we have already excluded the patients who took Cholrpromazine and cognitive enhancers for focusing on the effects of the intervention of this study. We have added this description in the exclusion criteria on Line 96.

4.5. Please report effect size formula.

Response 4.5: Thank you very much for the comments. The effect size of Cohen’s d for mean difference between groups defined by mean difference between two groups divided by square root (average of variances for two groups). The calculated effect size has been reported in Table 3. Please refer to Line 252 for the revision.

4.6. The GEE suggested SRLE effectiveness enough. The assumptions of ANCOVA are Normal distribution, equal variance and so on. Did you examine the assumptions of ANCOVA? To my opinion, keep GEE and delete the ANCOVA part.

Response 4.6: Thank you very much for the comments. The ANCOVA part has been deleted. The GEE model with the covariates of group, time, and group-by-time interaction for analysing the change from baseline of each score at months 3 and 6 was implemented to detect the occurrence of group-by-time interaction effect. The significant interaction effect indicated that the SRLE effect was different between months 3 and 6. The results were summarized in the upper part of Table 3. In the lower part of Table 3, the results of the comparison between SRLE and control groups at months 3 and 6 based on the change from baseline of each score by using the independent samples t-test were reported. Since all patient baseline and demographics had no significant difference, the independent samples t-test was used instead of ANCOVA in the revised manuscript. The assumption normality has been evaluated for implementation of the independent samples t-test. Please refer to Line 220 and 247 for the revision.

4.7. Please change age and duration of onset from categorical variables to continuous variables.

Response 4.7: Thank you for your suggestion. Age and duration of onset were changed as continuous variables and reanalysed accordingly. Please refer to Table 1 for the revision (Line 187). We have also renewed the statistical analysis and statements which were related to age and duration of onset in the Results section.

4.8. Please report IRB approve number.

Response 4.8: Thank you for your reminding. This study was approved by the Hospital Research Ethics Committee (REC No: IRB 102-170).  We reported the information on Line 97-98.

Point 5: Results

5.1. Please report effect sizes for SRLE on outcome indicators.

Response 5.1: The effect size of Cohen’s d for mean difference between groups was reported in Table 3. Please refer to Table 3 for the revision. (Line 252)

5.2. Table 1, please put age and duration of onset as continuous variables. The reason is: For 2 * 2table, it is generally recommended that there should be a minimum ‘expected’ frequency of 10 in each cell. If not, Fisher’s exact test should be used. No cell should contain zero observations.

Response 5.2: Thank you for the comment. Age and duration of onset were changed as continuous variables. The education level was realized by using the Fishers’ exact test. Please refer to Table 1 for the revision (Line 187).

5.3. Please change the p = 0.000 to p < .001, and report the third digit after the decimal point for all p values. It is not suitable for taking p values from third digit after the decimal point to rounding to two digit after the decimal point.

Response 5.3: All the p=0.000 has been changed as p < .001 accordingly.

5.4. Please combine Table 3 and Table 4 and report parameters of control variables and add B(SE) on the GEE model tables. The GEE model should report Group* T1 (EG vs. CG and baseline vs 3-month) and Group * T2 (EG vs. CG and baseline vs 6-month) interaction effect.

Response 5.4: Thank you for your suggestion. Table 3 and Table 4 have been combined in one Table named Table 3. The analyses of GEE were implemented for analysing the change from baseline of each score at month 3 and month 6 by considering the effects of group (SRLE vs. control), time (month 3 vs. month 6), and group-by-time interaction (Line 221-225). Both group and time were two levels of variables and SRLE and month 3 were chosen as the reference categories for group and time, respectively. Since both group and time are two levels of variables, only one interaction term was reported as their interaction effect. The information of reference categories have been added in the results of GEE of Table 3.

Point 6: Discussion

6.1. Please particularly attend to keeping the discussion of the paper and the conclusions within the bounds of our findings. That is, be careful not to overstate the significance or strength of the findings given the research design.

Response 6.1: Thank you for your reminding. We have reviewed the Discussion and Conclusion paragraphs and changed some descriptions to make it not to overstate the significance or strength of the findings. (Line 259, 308, 311, 318-320)

6.2. Not sure that it is appropriate to conclude that the SRLE may significantly enhance the cognitive function, improve the frontal lobe function, and reduce the BPSD in male adults with dementia aged over 65 years because of the research design limitations of the study, particularly potential bias due to authors assigned participants into experimental group or control group. It may be more appropriate to conclude that the intervention has potential and further study would be beneficial using RCT designs.

Response 6.2:

Thank you for your insightful recommendations. We have revised the conclusion as follow. “There is a potential bias due to it is not a random allocation to assign participants into the experimental group or control group, therefore, further study would be beneficial using RCT design and increase the number of participants and older female adults with dementia.” (Line 318-320)

Reviewer 2 Report

I am reviewing the paper entitled “Smart Restored by Learning Exercise Alleviates the Deterioration of Cognitive Function in Older Adults with Dementia” for the International Journal of Environmental Research and Public Health.  I really do not see any major flaws with the paper, except for the grammar and I will report on the problems that I found from beginning to the end of the paper.

The abstract should say “The results showed that the control group showed significantly lower…FAB scores than the SRLE group…”  The next sentence should say “The NPI scores were significantly higher in the control group than in the SRLE group…”  In the introduction, the authors should say “Because drugs for dementia treatment...side effects, behavior therapy with drugs has become…”  In the next paragraph, the authors should say “significantly correlating with cognitive functioning…”  On line 59, the authors should say “cognitive functions…”  On line 61, the authors should say “observation reflected that, without intervention…”  Line 62 should say “resulting in severe disabilities.”  Line 70 should say “on cognitive function”.  Line 91 should say “36 participants from”.  

Line 99 should say “The MMSE measure exhibits…and it is used to test…”  Line 100 should say “The test is comprised of eight subtests”.  Line 102 should say “High scores represent high levels of…”  Line 108 should say “High scores indicate great severity…”  Line 111 should say “The FAB showed…” and a period should end the sentence.  The reliability measure in the next sentence does not need a leading 0 for APA format.  Line 118 should say “The SRLE used in this study was comprised of two tasks: …” Then, the authors should list the two items and then start the next sentence as state.  Line 21 should say “The participants…read the Mandarin…materials were selected…”  Line 123 should say “…sentences of varying difficulty, including simple, medium, and complex.”  Line 130 should say “4 to 5 sheets”.  Line 131 should say “4 to 5 sheets”.  Line 132 should say “Approximately 20 to 30 minutes…”  Line 137 should say “group 4 to 5 days per week”. 

Line 143 should say “samples t-test”.  On line 144, the leading 0 should be removed.  Line 147-149 should say “36 adults in the experimental group and 32 adults in the control group.  Among the participants, 6 of them (8.8%)…65 to 80 years, and 62 of them (91.2%) were over 80 years old…”  Line 154 should say “The average MMSE pretest scores were lower for the experimental group (M = 14.33, SD = 4.39) than for the control group (M = 13.97, SD = 7.53).”  Line 156 should say “The BPSD measure was assessed…” Line 161 should replace the demonstrative “that” with “the average score”.  Line 174 and 175 should say “The FAB score for frontal lobe functioning increased from….”  Line 176 should say “the scores for the control group…”  Line 177 should say “The GEE measure…”  Line 180 should say “65 to 80 years” …80 years old.  Thus, age should …adjusted.  The GEE model included…age and it was…”  Line 190 should say “the score for the control group…lower than the scores for the experimental group…”  Line 192 should say “score for the control group…compared with the score for the experimental group…”  Line 193 and 194 should say “Therefore, the improvement in cognitive functioning for the experimental group was significantly greater than the improvement in cognitive functioning for the control group…”  The sentences on line 195 and line 197.  The same point is true on line 200 and line 202. 

Lines 209 and 210 should say “enhanced cognitive functioning…frontal love functioning”.  Line 216 should say “demonstrated that, after 6 months…”  Line 221 should say 2 to 6 days”.  Line 227 should say “5 days per week”.  Line 228 should say “8 to 10 weeks”.  Line 233 should say “In contrast to the study conducted by Kawashima, which indicated that relatively…”  Line 246 should say “consistent with the ones…”  Line 253 should replace “those” with “the ones”.  Line 261 should say “Our findings also showed that…”  Line 264 should say “all male individuals …80 years old.”  Line 268 should say “frontal lobe functioning”. 

Author Response

I am reviewing the paper entitled “Smart Restored by Learning Exercise Alleviates the Deterioration of Cognitive Function in Older Adults with Dementia” for the International Journal of Environmental Research and Public Health.  I really do not see any major flaws with the paper, except for the grammar and I will report on the problems that I found from beginning to the end of the paper.

Point 1: The abstract should say “The results showed that the control group showed significantly lower…FAB scores than the SRLE group…”  The next sentence should say “The NPI scores were significantly higher in the control group than in the SRLE group…”  In the introduction, the authors should say “Because drugs for dementia treatment...side effects, behavior therapy with drugs has become…”  In the next paragraph, the authors should say “significantly correlating with cognitive functioning…”  On line 59, the authors should say “cognitive functions…”  On line 61, the authors should say “observation reflected that, without intervention…”  Line 62 should say “resulting in severe disabilities.”  Line 70 should say “on cognitive function”.  Line 91 should say “36 participants from”. 

Response 1:

Thank you for your kindly advices on this study. Your suggestions make our manuscript more solid and readable. We have completed the corrections one by one according to your suggestions. In the revision, the age and duration of onset were changed as continuous variables and reanalysed accordingly. We have also renewed the statistical analysis which was related to age and duration of onset in the Results section. Thanks again for your help.

Point 2: Line 99 should say “The MMSE measure exhibits…and it is used to test…”  Line 100 should say “The test is comprised of eight subtests”.  Line 102 should say “High scores represent high levels of…”  Line 108 should say “High scores indicate great severity…”  Line 111 should say “The FAB showed…” and a period should end the sentence.  The reliability measure in the next sentence does not need a leading 0 for APA format.  Line 118 should say “The SRLE used in this study was comprised of two tasks: …” Then, the authors should list the two items and then start the next sentence as state.  Line 21 should say “The participants…read the Mandarin…materials were selected…”  Line 123 should say “…sentences of varying difficulty, including simple, medium, and complex.”  Line 130 should say “4 to 5 sheets”.  Line 131 should say “4 to 5 sheets”.  Line 132 should say “Approximately 20 to 30 minutes…”  Line 137 should say “group 4 to 5 days per week”.

Response 2: We have completed the corrections one by one according to your suggestions.

Point 3: Line 143 should say “samples t-test”.  On line 144, the leading 0 should be removed.  Line 147-149 should say “36 adults in the experimental group and 32 adults in the control group.  Among the participants, 6 of them (8.8%)…65 to 80 years, and 62 of them (91.2%) were over 80 years old…”  Line 154 should say “The average MMSE pretest scores were lower for the experimental group (M = 14.33, SD = 4.39) than for the control group (M = 13.97, SD = 7.53).”  Line 156 should say “The BPSD measure was assessed…” Line 161 should replace the demonstrative “that” with “the average score”.  Line 174 and 175 should say “The FAB score for frontal lobe functioning increased from….”  Line 176 should say “the scores for the control group…”  Line 177 should say “The GEE measure…”  Line 180 should say “65 to 80 years” …80 years old.  Thus, age should …adjusted.  The GEE model included…age and it was…”  Line 190 should say “the score for the control group…lower than the scores for the experimental group…”  Line 192 should say “score for the control group…compared with the score for the experimental group…”  Line 193 and 194 should say “Therefore, the improvement in cognitive functioning for the experimental group was significantly greater than the improvement in cognitive functioning for the control group…”  The sentences on line 195 and line 197.  The same point is true on line 200 and line 202.

Response 3: We have completed the corrections one by one according to your suggestions.

Point 4: Lines 209 and 210 should say “enhanced cognitive functioning…frontal love functioning”.  Line 216 should say “demonstrated that, after 6 months…”  Line 221 should say 2 to 6 days”.  Line 227 should say “5 days per week”.  Line 228 should say “8 to 10 weeks.  Line 233 should say “In contrast to the study conducted by Kawashima, which indicated that relatively…”  Line 246 should say “consistent with the ones…”  Line 253 should replace “those” with “the ones”.  Line 261 should say “Our findings also showed that…”  Line 264 should say “all male individuals …80 years old.”  Line 268 should say “frontal lobe functioning”.

Response 4: We have completed the corrections one by one according to your suggestions.

Reviewer 3 Report

The main purpose of this study is to see the effects of SRLE on cognitive function, neuropsychiatric symptoms, and frontal lobe function among older adults with dementia. It seems that the results shows apparent effectiveness of SRLE but some methodological revision is needed to clarify the results.

Comment 1:

In Table 1, author needs to explain why age is used as categorical variable. I think it should be continuous variable.

Comment 2:

(Table 1) Why there is no middle school in Education? Also, it is better if author treats education as continuous variable (e.g., years of education).

Comment 3:

(Table 1) Again, onset should also be treated as continuous variable.

Comment 4:

(Table 1) The total sample size of intervention group is 33, not 36. And the total sample size of control group is 35, not 32.

Comment 5:

(Table 1) No gender information?

Comment 6:

In Table 2, the models should at least adjust for age. Author can show both adjusted model and unadjusted model in the table, so readers can compare those two.

Comment 7:

In line 182 and 183, author mentioned that the time of onset and age exhibited potential interaction in Table 3. However, I don’t see any potential interaction b/w onset and age in Table 3. Please explain. I think only age needs to be controlled (because of sample size and consistency with other tables).

Comment 8:

What is group-by-time-age interaction? (In line 184) Please explain.

Comment 9:

In Table 3, baseline does not need to be controlled because there were no significant differences in baseline values (MMSE, NPI, and FAB) between intervention and control group.

Comment 10:

Please mention control variables in Table 3.

Comment 11:

Again, in Table 4, baseline does not need to be controlled.

Author Response

The main purpose of this study is to see the effects of SRLE on cognitive function, neuropsychiatric symptoms, and frontal lobe function among older adults with dementia. It seems that the results shows apparent effectiveness of SRLE but some methodological revision is needed to clarify the results.

Point 1: In Table 1, author needs to explain why age is used as categorical variable. I think it should be continuous variable.

Response 1: Thank you for your kindly advices on this study. Your suggestions make our manuscript more solid and readable. The age has been revised as a continuous variable, and we have also renewed the statistical analysis which was related to age in the Results section.

Point 2: (Table 1) Why there is no middle school in Education? Also, it is better if author treats education as continuous variable (e.g., years of education).

Response 2: Thank you for your reminding. We made a mistake in the terminology of education; we have revised elementary school to middle school. We still use education as a categorical variable as recommended by other reviewers.

Point 3: (Table 1) Again, onset should also be treated as continuous variable.

Response 3: We agree your opinion. The onset has been revised as a continuous variable, and we have also renewed the statistical analysis which was related to the onset.

Point 4: (Table 1) The total sample size of intervention group is 33, not 36. And the total sample size of control group is 35, not 32.

Response 4: We have revised the errors in Table 1. Thank you very much.

Point 5: (Table 1) No gender information?

Response 5: Because all participants are male veterans in this study, so there’s no gender information in Table 1. We have mentioned the gender information on Line 179.

Point 6: In Table 2, the models should at least adjust for age. Author can show both adjusted model and unadjusted model in the table, so readers can compare those two.

Response 6: The age has been changed to a continuous variable, and there was no significant difference between two groups, so, the model didn’t adjust for age in Table 2.

Point 7: In line 182 and 183, author mentioned that the time of onset and age exhibited potential interaction in Table 3. However, I don’t see any potential interaction b/w onset and age in Table 3. Please explain. I think only age needs to be controlled (because of sample size and consistency with other tables).

Response 7: Thank you for your reminding. It’s our typo. Only the group-by-time interaction effect was included in GEE model. The related description has been removed and corrected. Please refer to Line 220 and 233 for the correction.

Point 8: What is group-by-time-age interaction? (In line 184) Please explain.

Response 8: It’s our typo. Only the group-by-time interaction effect was included in GEE model. The related description has been removed and corrected. Thank you very much.

Point 9:In Table 3, baseline does not need to be controlled because there were no significant differences in baseline values (MMSE, NPI, and FAB) between intervention and control group.

Response 9: The baseline effect has been removed from GEE model. Also, since the baseline effect is unnecessary to be considered, the independent sample t-test was implemented instead of ANCOVA for analysing the SRLE effect at month 3 and month 6, respectively. Please refer to Table 3 for the revision.

Point 10: Please mention control variables in Table 3.

Response 10: In GEE model, there were three variables (effects) including group, time, group-by-time effects included for analysing the change from baseline of each score at months 3 and 6. SRLE and month 3 were chosen as the reference categories for group and time effect, respectively. The information of reference categories have been added in the text and please refer to Table 3 for the revision.

Point 11: Again, in Table 4, baseline does not need to be controlled.

Response 11: Thank you very much for the comments. The baseline effect has been removed from GEE model. Also, since the baseline effect is unnecessary to be considered, the independent sample t-test was implemented instead of ANCOVA for analysing the SRLE effect at month 3 and month 6, respectively. Table 3 and Table 4 have been combined as one Table named Table 3. Please refer to Table 3 for the revision.

Round 2

Reviewer 1 Report

Manuscript No: ijerph-455326-V2

Title: Smart Restored by Learning Exercise Alleviates the 2 Deterioration of Cognitive Function in Older Adults 3 with Dementia: A quasi-experimental research

Title:

A quasi-experimental research not A quaxi-experimental research

Measures’ copyright:

I just remind authors that the MMSE is not free, the copyright belong to PAC inc.

https://www.parinc.com/products/pkey/237?fbclid=IwAR2D3UqOH9m1CXmMWYnCQAGCY4xyUdOa57MEZvdDGpGXFx0qVh-lYOHDQUs

Please describe this part.

Methods:

The data collector is the same as the intervener. The intervener knew who the experimental group members are. This is a major flaw of research design. This flaw may impact on post-test data and research results’ inference. Please add this flaw into discussion section.

The experimental program was an individualized strategy instead of a group intervention. Please add into manuscript.

Table

Table 1: Fisher’s exact test only suits for 2*2 table. It is required that each cell be non-zero for Fisher’s exact test. I suggest that education be regarded as a continuous variable (year of education). E.g., none as 0, primary school as 6, secondary school as 9, …etc.

Please classify the meanings of data on GEE table, and then you can understand lower part of table 3 is redundant. T-test is a parameter statistics.

The assumption of normality was checked by Wilk-Saphiro test. Please provide detailed results of Wilk-Saphiro test.

Table 3

        Table 3 report wasn’t a standardized GEE table. Please see the attached file.

Author Response

please refer to the attachment file, thank you.

Reviewer 2 Report

I am reviewing the paper entitled “Smart Restored by Learning Exercise Alleviates the Deterioration of Cognitive Function in Older Adults with Dementia – A Quasi-Experimental Research” for the International Journal of Environmental Research and Public Health should say “A Quasi Experiment” or “Quasi-Experimental Research”.  Like last time, most of my comments pertain to English.   

The abstract should say “The Group by Time interaction was statistically significant for MMSE and FAB scores, which”.  Line 38 starts a sentence with the acronym “BPSD” and sentences should not start with acronym.  Line 39 should say “symptoms, such as”.  Line 41 talks about a systematic review of two meta-analyses and I am not sure to which study the authors are referring.  The authors should either cite a reference.  Line 45 seems to put several spaces after “Because”.  Line 46 seems to put several spaces before “behavior therapy” and a space is needed between “drugs” and “has”.  Line 58 should say “Frontal lobe degeneration significantly correlated with cognitive functions, such as memory impairment, declines in executive functions,”.  Line 64 should say “randomized experiment…76 to 86 years.”  Line 76 should say “were conducted 4 to 5 days”.  Line 77 should say “effects on cognitive functions”.  Line 92 should say “those individuals who took”.  Line 117 should say “High scores indicate great severity”.  On line 168, the t-values should be italicized, no spaces should surround the hyphens and it should say “paired-samples” and “independent samples t-test”.  Line 170 should surround the less than sign with spaces.  Line 179 uses two periods back to back.  Line 195 should say “symptoms for the SRLE group”.  Line 197 should say “the scores for the control group”.  Line 198 should say “symptoms improved favorably for participants in the SRLE group than for participants in the control group.”  Line 204 should say “frontal lobe functioning”.  Line 207 starts a sentence “It” and I have no idea what it is.  Line 208 should say “with an increasing trend”.  Line 210 should italicize the “t” in t-test.  Line 218 should say “significantly lower” rather than “significantly less” and Line 224 should say “frontal lobe functioning”.  Line 235 should say “calculations can effectively enhance cognitive functioning, mitigate BPSD and improve frontal lobe functioning”.  Line 237 should say “group increased with the length of intervention employment.”  Line 242 should say “Previous randomized experiments”.  Line 245 should say “cognitive functioning”.  Line 248 should say it twice.  Line 254 should separate “to” and “10”.  Lines 255 and 258 should say “cognitive functioning”.  Line 256 should say “and, thereby, prevent”.  Line 272 should say “consistent with the ones in the”.  Line 280 should say “non-drug”.  Line 282 should move the top right part of line 281 down to the next line.  Line 283 should say “The present study explained that the SRLE program potentially reduces the decline in cognitive functioning and frontal lobe functions were”.  Line 284 should say “Our findings showed that”.  Line 285 should say “However, the study contained limitations.”  Line 289 should say “The current study is potentially biased because participants were not randomly assigned to the experiment or control groups.  Therefore, further study would benefit from using RCT designs and increasing …”

Author Response

We have completed the corrections one by one according to your suggestions. Please refer to the revision file. Your suggestions make our manuscript more understandable. Thank you for your advice on this study.

Reviewer 3 Report

The quality of this research is improved. If I add just one more comment, it is better if time is considered as continuous variable. But this is author's choice.

Author Response

Thanks for your encouragement. In the revision, the education was changed as continuous variable and re-analysed accordingly. We have also renewed the statistical analysis which was related to education in the Results section. Thanks again for your help.